# Clinical Features and Outcomes of Conversion Therapy in Patients with Unresectable Hepatocellular Carcinoma

**DOI:** 10.3390/cancers15215221

**Published:** 2023-10-30

**Authors:** Tetsu Tomonari, Joji Tani, Yasushi Sato, Hironori Tanaka, Takahiro Tanaka, Tatsuya Taniguchi, Yutaka Kawano, Asahiro Morishita, Koichi Okamoto, Masahiro Sogabe, Hiroshi Miyamoto, Tsutomu Masaki, Tetsuji Takayama

**Affiliations:** 1Department of Gastroenterology and Oncology, Institute of Biomedical Sciences, Tokushima University Graduate School of Medicine, Tokushima 770-8504, Japan; tetsu.tomonari@gmail.com (T.T.); tanaka.hironori@tokushima-u.ac.jp (H.T.); tanaka_takahiro2002@yahoo.co.jp (T.T.); qyrwx456@yahoo.co.jp (T.T.); ykawano@tokushima-u.ac.jp (Y.K.); okamoto.koichi@tokushima-u.ac.jp (K.O.); sogabe.masahiro@tokushima-u.ac.jp (M.S.); miyamoto.hiroshi@tokushima-u.ac.jp (H.M.); takayama@tokushima-u.ac.jp (T.T.); 2Department of Gastroenterology and Neurology, Kagawa University Graduate School of Medicine, Kagawa 761-0701, Japan; georget@med.kagawa-u.ac.jp (J.T.); asahiro@med.kagawa-u.ac.jp (A.M.); tmasaki@med.kagawa-u.ac.jp (T.M.); 3Department of Community Medicine for Gastroenterology and Oncology, Institute of Biomedical Sciences, Tokushima University Graduate School of Medicine, Tokushima 770-8503, Japan

**Keywords:** hepatocellular carcinoma, lenvatinib, atezolizumab, bevacizumab

## Abstract

**Simple Summary:**

Conversion therapy has shown potential for improving the prognosis of patients with unresectable hepatocellular carcinoma (u-HCC). However, information on the characteristics and outcomes of patients undergoing conversion therapy is lacking. We examined 244 patients with u-HCC treated with lenvatinib (LEN) and atezolizumab + bevacizumab (Atezo + Bev). Of the 244 patients, 12 (4.9%) underwent conversion therapy, six out of 131 (4.6%) were treated with LEN, and six out of 113 (5.3%) were treated with Atezo + Bev. Eleven patients (91.7%) with a modified albumin bilirubin (mALBI) grade 1 or 2a and BCLC-B stage showed significantly higher rates of transition to conversion therapy (*p* < 0.05). Among the patients with u-HCC who were treated with LEN and Atezo + Bev, those with mALBI 1+2a and BCLC-B were likely to achieve conversion therapy with downstaging. The outcomes of the patients undergoing conversion therapy are promising.

**Abstract:**

This retrospective multicenter study analyzed 244 patients with unresectable hepatocellular carcinoma treated with lenvatinib (LEN) and atezolizumab + bevacizumab (Atezo + Bev) to examine the characteristics, treatment courses, and prognoses. The cases of patients who could achieve HCC downstaging from Barcelona Clinic Liver Cancer (BCLC) stage B or C to A or zero indicated the need for conversion therapy. The patients’ prognoses with and without conversion therapy were compared. Of the 244 patients, 12 (4.9%) underwent conversion therapy, six out of 131 (4.6%) were treated with LEN, and six out of 113 (5.3%) were treated with Atezo + Bev. Eleven patients (91.7%) with a modified albumin bilirubin (mALBI) grade 1 or 2a and BCLC-B stage showed significantly higher rates of transition during conversion therapy (*p* < 0.05). The patients undergoing conversion therapy had a significantly longer median overall survival rate than those receiving chemotherapy alone (1208 [1064–NA] vs. 569 [466–704] days, *p* < 0.01). A comparison of the patients who achieved a partial response with and without conversion was evaluated using propensity score matching to reduce the confounding factors, showing a significant survival benefit in the conversion group (1208 [1064–NA] vs. 665 days, *p* < 0.01). Among the patients with u-HCC who were treated with LEN and Atezo + Bev, those with mALBI 1 + 2a and BCLC-B were likely to achieve conversion therapy with downstaging.

## 1. Introduction

Hepatocellular carcinoma (HCC) is the leading cause of cancer death. Therefore, there is an urgent need to develop a cure [1,2]. Recently, remarkable progress has been made in developing drug therapy for unresectable hepatocellular carcinoma (u-HCC), and the efficacy of molecularly targeted agents (MTAs) and immune checkpoint inhibitors have been reported. 

Currently, six different regimens are approved in Japan, and agents with high response rates are being approved and used. First, a recent phase III REFLECT trial indicated that lenvatinib (LEN) was non-inferior to sorafenib (SOR) as a first-line treatment for u-HCC (median overall survival [OS], 13.6 vs. 12.3 months; hazard ratio [HR], 0.92; 95% confidence [CI], 0.79–1.06) [3]. LEN is an oral MTA that targets the vascular endothelial growth factor (VEGF) receptors 1–3, fibroblast growth factor receptors 1–4, platelet-derived growth factor receptor α, RET, and KIT [4,5,6,7,8] and is characterized by a high response rate of 40.6% and 18.8% for a modified Response Evaluation Criteria in Solid Tumors (mRECIST) evaluation and RECIST version 1.1, respectively.

Second, the phase III IMbrave150 clinical trial demonstrated that the combined therapy of atezolizumab (a monoclonal antibody targeting the programmed cell death ligand 1) and bevacizumab (a monoclonal antibody for VEGF A) (Atezo + Bev) was superior to SOR alone as a first-line treatment for u-HCC (median OS, not reached vs. 13.2 months; HR, 0.58; 95% CI, 0.42–0.79) [9]. The results of this phase III trial established Atezo + Bev therapy as the current first-line therapy for u-HCC [10,11]. Additionally, this combination therapy was similar to LEN therapy, characterized by high response rates of 33.2% and 27.3% for the mRECIST evaluation and RECIST version 1.1, respectively.

These two regimens have been used in clinical practice in many u-HCC cases, and their efficacies have been reported [12,13,14]. Among them, there have been recently reported successful cases of conversion therapy where the cases achieved a high response and downstaging when treated with additional treatments, such as hepatectomy or ablation aimed at a cure [15,16,17].

However, no reports exist detailing the characteristics and prognoses of patients with u-HCC who achieved conversion therapy with LEN or Atezo + Bev. Therefore, to clarify the characteristics of the patients with u-HCC who achieved conversion therapy with LEN or Atezo + Bev, we retrospectively analyzed their clinical features and outcomes.

## 2. Materials and Methods

### 2.1. Patient Selection and HCC Diagnosis 

This retrospective observational study evaluated the efficacy of LEN therapy (Eisai Pharmaceutical Co., Ltd., Tokyo, Japan) and Atezo + Bev (Chugai Pharmaceutical Co., Ltd., Tokyo, Japan) combination therapy in patients with u-HCC who were treated at the Tokushima and Kagawa University Hospitals between March 2018 and December 2022. The inclusion criteria were based on the REFLECT and IMbrave150 studies [3,9]. Briefly, eligible patients had evaluable lesions based on the RECIST [18] and the mRECIST [19] criteria, an Eastern Cooperative Oncology Group performance status (ECOG-PS) score of zero or one point [20], Barcelona Clinic Liver Cancer (BCLC) stage B or C [21], and Child–Pugh (CP) class A. In LEN therapy, the initial dose was reduced from 12 to 8 mg once daily for the patients with CP class B, as described in the clinical phase I study [22]. In Atezo + Bev therapy, the patients with CP class B were treated according to the recommendation of the GO30140 study [23]. For the patients with BCLC-B, drug therapy was initiated for the transcatheter arterial chemoembolization (TACE)-refractory or TACE-unsuitable cases [10,24]. The HCC diagnosis was based on the guidelines established by the Liver Cancer Study Group of Japan [15]. According to these guidelines, HCC is diagnosed based on histological or characteristic radiological findings. For example, in portal-phase or equilibrium-phase images obtained using dynamic computed tomography (CT) or contrast-enhanced magnetic resonance imaging (MRI), the typical arterial enhancement of the tumor is followed by a washout pattern. 

This study protocol was approved by the ethics committee of Tokushima University Hospital (approval number: 3816) and each participating institution. This study adhered to the guidelines of the 1975 Declaration of Helsinki. 

### 2.2. Treatment Protocol

LEN was administered orally daily at 12 mg and 8 mg for the patients weighing ≥60 kg and ≤60 kg, respectively. If an adverse event of a grade ≥ 3 occurred, the drug was withdrawn until recovery was achieved.

For the Atezo + Bev therapy, the patients received 1200 mg and 15 mg/kg of atezolizumab and bevacizumab intravenously every 3 weeks, respectively. When serious adverse events (AEs) were observed, including grade 3 AEs or unacceptable grade 2 AEs, the Atezo + Bev therapy was discontinued until the patient recovered from the AEs and reverted to a lower grade. 

### 2.3. Patient Outcomes

The patients were observed for at least 4 weeks. Safety was evaluated by recording the side effects, clinical laboratory findings, physical findings, vital sign measurements, hematological and biochemical tests, and urinalysis. The radiological response to the treatment was assessed 6–8 weeks after initiating the LEN or Atezo + Bev therapy and every 6–8 weeks thereafter based on RECIST and mRECIST. The objective response rate (ORR) was defined as the sum of the complete responses (CRs) and partial responses (PRs). The disease control rate (DCR) was defined as the sum of the CR, PR, and stable disease (SD) rates. Furthermore, progression-free survival (PFS) was defined as the time from the first day of the LEN or Atezo + Bev therapy to the confirmation of radiographic progression or death from any cause. In most cases, we established specific criteria for initiating conversion therapy in the patients who had the potential to achieve downstaging of HCC from BCLC stage B or C to A or zero. The choice of treatment for conversion therapy hinged on several factors, including the patient’s general condition, hepatic reserve function, tumor size, tumor number, and the precise localization of HCC. These decisions were made collaboratively, following consultation with our liver surgeon. The ablation procedures were predominantly conducted using microwave ablation or radiofrequency ablation. Case 6, however, deviated from this typical approach. Despite being classified as a BCLC-A case, we initiated treatment with LEN as an exception. This decision was rooted in the patient’s inoperable status, primarily due to an elevated ICG value of 47%. Subsequently, we discontinued LEN therapy due to adverse events and opted for hepatic arterial embolization after the patient’s recovery. Following this, microwave ablation therapy was performed as an additional radical treatment.

### 2.4. Hepatic Functional Reserve

The liver functional reserve was assessed according to the CP classification system and mALBI grade. The mALBI grades were assigned based on the serum albumin and total bilirubin levels using the following formula. ALBI score = (log_10_ bilirubin [µmol/L] × 0.66) + (albumin [g/L] × −0.085). The formula was defined using the following scores: −2.60 = grade 1, >−2.60 to ≤−2.27 = grade 2a, >−2.27 to ≤−1.39 = grade 2b, and >−1.39 = grade 3 [20].

### 2.5. Statistical Analysis 

The categorical variables were compared using Fischer’s exact test, and the continuous variables were compared using the Mann–Whitney U and Kruskal–Wallis tests. All the significance tests were two-sided, and a statistical significance was set as *p* < 0.05. Furthermore, the PFS and OS were analyzed using the Kaplan–Meier curve with the log-rank test. (with 95% CI). All the statistical analyses were performed using the Easy R version 1.29 software (Jichi Medical University Saitama Medical Center, Saitama, Japan) [25]. A propensity score analysis was performed to reduce any bias affecting the patient selection and to investigate the association between treatment and outcome. For the propensity score matching cohort, the propensity score matching criteria were adjusted for age, sex, CP score, ECOG-PS, and portal vein invasion (Vp) when drug therapy was initiated.

## 3. Results

### 3.1. Patient Characteristics

Table 1 summarizes the baseline patient background of the study population. We administered LEN or Atezo + Bev to 266 patients between March 2018 and December 2022. Twenty-two cases were excluded from the analysis because of the difficulty in the imaging evaluation using contrast-enhanced CT or MRI. The median age of the patients was 73 years (quartiles, 67–79 years) and included 48 (19.6%) women. The ECOG-PS was zero in 189 (77.5%) patients. Of the 244 patients, 32 (13.1%) and 94 (38.5%) were hepatitis B virus and hepatitis C virus (HCV) antibody positive, respectively. The CP scores before drug initiation were five, six, seven, and eight points in 142, 80, 16, and six patients, respectively. The mALBI grades before drug initiation were 1, 2a, 2b, and 3 points in 84, 66, 90, and six patients, respectively. Additionally, the median alpha-fetoprotein (AFP) level was 87 ng/mL (quartiles 8–1,013 ng/mL), and systemic therapies were initiated at BCLC stages B and C in 127 and 117 patients, respectively. Among the 244 patients, 156, 51, 25, eight, and four were treated with MTA-naive (first line), second-line, third-line, fourth-line, and fifth-line treatments, respectively. There was no significant difference in the baseline characteristics of the patients treated with Atezo + Bev (113 patients) and those treated with LEN (131 patients), except for the treatment line.

### 3.2. Treatment Effect

The median observation periods for the Atezo + Bev and LEN therapies were 371 (quartiles 203–604) days and 388 (quartiles 203–604), respectively. The results of the antitumor efficacy evaluation using RECIST version 1.1 are presented in Appendix A. The antitumor efficacy of Atezo + Bev showed that among the 113 patients, one (0.8%), 22 (19.5%), 67 (59.3%), and 23 (20.4%) had CR, PR, SD, and progressive disease (PD), respectively. The ORR and DCR were 20.4% and 79.6%, respectively. The antitumor efficacy of LEN showed that among the 131 patients, two (1.5%), 33 (25.1%), 82 (62.6%), and 14 (10.7%) had CR, PR, SD, and PD, respectively. The ORR and DCR were 26.7% and 89.3%, respectively (Appendix A). The median PFS for Atezo + Bev and LEN, evaluated using the RECIST version 1.1, were 6.1 and 6.2 months, respectively (Appendix A). Furthermore, the median OS for Atezo + Bev and LEN were 26.8 and 18.6 months, respectively (Appendix A).

### 3.3. Baseline Characteristics of the Conversion Cases

Of the 244 patients, 12 (4.9%) underwent conversion therapy, six out of 113 (5.3%) were treated with Atezo + Bev, and six out of 131 (4.6%) were treated with LEN (Table 2). The median age of the 12 conversion cases was 73 years (quartiles 71–78 years) and included four women. Five patients were HCV-Ab positive and seven had NASH or alcoholic hepatitis. Except for Case 4, 11 (91.7%) were initiated using first-line treatment and had a median AFP level of 13 (quartiles 5–409) ng/mL. The liver reserve was Child–Pugh class A in all the cases (100%). The Child–Pugh class scored five in nine cases (75%), and six in three (25%) cases. The mALBI grade was 1 or 2a in 11 cases (91.7%). There were no significant differences in the characteristics of the patients who underwent conversion therapy between the two groups. The median tumor number and size were 4 (quartiles 3–5) and 47 mm (quartiles 31–65), respectively. In total, 11 cases were BCLC-B (91.7%) and one (8.3%) was BCLC-C (Appendix A). Furthermore, a univariate analysis was performed to analyze the factors involved in the transition during conversion therapy. The factors included the age, sex, etiology, ECOG-PS, mALBI grade, Child–Pugh score, BCLC stage, AFP level, and drug therapy. The results showed significantly higher rates of conversion in patients with mALBI1+2a (*p* = 0.03) and BCLC-B (*p* = 0.01) (Table 3). Furthermore, it was noteworthy that within the LEN group, a higher rate of conversions during conversion therapy was observed in the BCLC-B cases (Appendix A). Conversely, in the Atezo + Bev group, a significantly greater proportion of conversions during conversion therapy was observed among the patients with mALBI-1+2a (Appendix A). 

### 3.4. Changes in Tumor Factors in Conversion Cases

Changes in the tumor factors from the baseline in the cases that led to conversion therapy are presented in Table 4, and the treatment courses of the conversion cases are shown in Figure 1. In the conversion therapy cases, all the patients had a PR according to the RECIST version 1.1 evaluation. Conversion therapy was performed in cases where a decrease in the tumor number or maximum tumor diameter was obtained, resulting in downstaging to BCLC-A or BCLC-0. In Case 7, portal vein invasion persisted, and the patient was consulted by a hepatobiliary and pancreatic surgeon, who determined that the patient was technically eligible for hepatectomy. Therefore, conversion surgery was performed in Case 7. When hepatic resection became feasible, conversion by hepatectomy was performed. Changes in the tumor factors before conversion therapy included the following. The median AFP changed from 88 ng/mL (quartiles 6–1033) to 7 ng/mL (quartiles 5–12), the median number of tumors changed from four (quartiles 3–5) to one (quartiles 1–2), and the median maximum tumor diameter changed from 47 mm (quartiles 31–65) to 25 mm (quartiles 14–35). Additionally, the median treatment duration and interruption were 211 days (quartiles 162–310) and 28 days (quartiles 10–35), respectively, before conversion therapy. Among these cases, hepatectomy was performed for those with the largest tumor diameter exceeding 3 cm, whereas ablation therapy was performed for lesions < 3 cm.

### 3.5. Long-Term Outcomes of the Post-Conversion Case

In total, eight of the 12 cases relapsed, and the median relapse-free survival time was 378 days (quartiles 197–476 days; Table 5). Of these patients, five of the 12 (41.7%) continued drug therapy after conversion therapy. However, no significant difference was found in the recurrence-free survival (RFS) between the continuation and non-continuation groups (drug therapy—vs. drug therapy + 447 days [157–NA] vs. 483 days [239–NA], *p* = 0.88; Appendix A). The comparison of the RFS between the group treated with ablation therapy during conversion therapy and the liver resection group showed a slight trend toward a longer RFS in the liver resection group, but the difference was not significant (ablation group, 274 days [157–NA]; hepatectomy group, NA days [447–NA]; *p* = 0.09; Appendix A). Additionally, the median OS was significantly longer in the patients who underwent conversion therapy (conversion *−* 569 days [466–704] conversion + 1208 days [1064–NA] *p* < 0.01; Figure 2). No complications were found after conversion therapy, including surgical resection or ablation therapy, in these 12 patients. Furthermore, a comparison of the patients with a PR in RECIST version 1.1 (n = 55) with and without conversion was conducted using a propensity score matching analysis (conversion + n = 10 vs. conversion *−* n = 10; Appendix A). In the comparison of the patient backgrounds between the conversion and non-conversion groups, no significant differences were observed (Appendix A). In addition, the OS was compared between the conversion and non-conversion groups, and a significant survival advantage was observed in the conversion group (conversion + 1208 days [1064–NA] vs. conversion *−* 665 days [89–NA]; *p* < 0.01; Figure 3).

## 4. Discussion

In this study, 12 of the 244 (4.9%) patients with u-HCC underwent conversion therapy after receiving LEN or Atezo + Bev treatment. The characteristics of these patients included mALBI1+2a and BCLC-B. The patients who received conversion therapy had significantly better long-term survival, and no significant difference was found in the treatment choice between ablation therapy and hepatic resection. 

Since the recent approval of LEN and Atezo + Bev, there have been a few case reports on successful cases of conversion therapy owing to their high response rates [15,26,27]. However, there have been few coherent reports, and no study has mentioned what kinds of cases can be converted during conversion therapy. 

In a comprehensive report, Kudo et al. reported a 35% conversion rate with Atezo + Bev in patients with BCLC-B who were TACE-unsuitable [17]. In contrast, the conversion rates for Atezo + Bev cases in our study tended to be low (5.3%; overall, 4.9%). We believe that the reason for this difference was that in the study by Kudo et al., conversion therapy was also given to patients with SD or PD when additional treatment was considered effective, whereas in our case, additional therapy was given only when downstaging of HCC from BCLC stage B or C to A or 0 was achieved, even in patients with PR. Another reason was that Kudo et al. considered super-selective TACE as a treatment option in addition to ablation and hepatectomy during conversion therapy. However, in this study, ablation and hepatectomy were the only curative treatments.

Long-term results from using conversion therapy in liver cancer drug therapy have been reported with SOR, and additional liver resection during SOR treatment has been a significant factor for achieving survival of >3 years [28]. However, it has a lower response rate than LEN or Atezo + Bev [3,9,29]. Therefore, conversion therapy using LEN or Atezo + Bev is expected to become a mainstream treatment. 

Regarding the cases where conversion therapy was achieved, most cases in this study were first-line or mALBI 1 + 2a. Similarly, it has been reported that an early treatment line and good hepatic reserve are associated with a good therapeutic response when using LEN [30,31,32] and Atezo + Bev [13,33,34]. Thus, starting chemotherapy as early as possible with a good hepatic reserve may increase the likelihood of achieving conversion therapy.

Additionally, although the guidelines state that the standard care for BCLC stage B is TACE, systemic therapy is recommended as the next treatment option for TACE-refractory cases [10,24]. It is recommended that upfront systemic therapy should be the preferred treatment strategy in TACE-unsuitable cases with BCLC stage B [35]. In our study, most of the patients who could undergo conversion therapy were those who started drug therapy as BCLC-B TACE-unsuitable patients, suggesting that the introduction of systemic therapy at BCLC-B can lead to curative treatment.

Since the angiogenesis inhibitory effects of LEN and bevacizumab can cause complications, such as delayed healing and bleeding, treatment should be discontinued preoperatively if surgery or ablation procedures are required. However, no established protocol exists for the appropriate timing of the discontinuation of LEN and bevacizumab. Since the half-life of LEN is approximately 28–35 h [21], we reported that the interruption period before surgery or ablation should be at least 1 week, given the safety margin [15]. Additionally, several studies on the withdrawal period of bevacizumab have been reported in conversion therapy for colorectal cancer, and we determined that a withdrawal period of at least 28 days was appropriate [36]. 

The results of the STORM trial were negative for the efficacy of adjuvant therapy in HCC [22]. However, it was reported that ablation therapy for early-stage HCC and Atezo + Bev therapy for post-hepatectomy cases prolonged RFS, and it is expected to be an adjuvant therapy in the future [37]. Conversely, the efficacy of adjuvant therapy in patients who underwent conversion therapy at the intermediate or advanced stages is unknown. In our study, adjuvant therapy was administered in five cases (three LEN and two Atezo + Bev cases). However, no difference was observed in the RFS between the patients treated with and without adjuvant therapy. Therefore, given the small number of cases, further prospective studies are needed to determine whether adjuvant therapy is necessary after conversion. Furthermore, the possibility of the influence of underlying therapy on the post-treatment course has also been reported, and a further accumulation of cases is needed in the future [38].

Kudo et al. reported that liver resection, ablation therapy, and selective TACE were treatment options for conversion therapy using Atezo + Bev therapy, with good results [17]. In our study, liver resection and ablation therapy were selected as curative treatment options, and both treatments had good long-term results, with no significant differences in post-treatment complications or RFS. These results suggest that both treatment modalities are options for conversion therapy, subject to close consultation between the internists and surgeons. On the other hand, in patients with hepatocellular carcinoma after radiofrequency ablation, previous studies have highlighted the significance of the clinical and tumor parameters evaluated at the time of recurrence, especially the type of recurrence pattern, for influencing survival after recurrence. Therefore, the evaluation at the time of recurrence may hold an equal importance in conversion cases [39].

The main limitations of our study were its retrospective nature, small sample size, and short observational period. Therefore, a large-scale prospective study is required to confirm our findings and conduct more detailed analyses. In addition, although the conversion group demonstrated a significant extension in the overall survival, it is essential to acknowledge the presence of immortal time within this group. It is plausible that immortal time bias may have influenced the observed survival outcomes in this study.

## 5. Conclusions

In conclusion, these results demonstrated that LEN and Atezo + Bev may increase the possibility of conversion therapy in patients with u-HCC who exhibit a good hepatic reserve function and BCLC-B status, and that these patients may have a favorable prognosis.

## Figures and Tables

**Figure 1 cancers-15-05221-f001:**
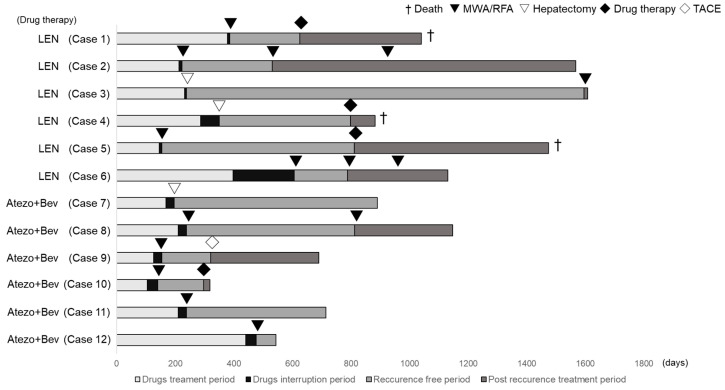
Treatment course of the conversion cases. Cases 1–6 were treated with lenvatinib, and Cases 7–12 were treated with atezolizumab + bevacizumab.

**Figure 2 cancers-15-05221-f002:**
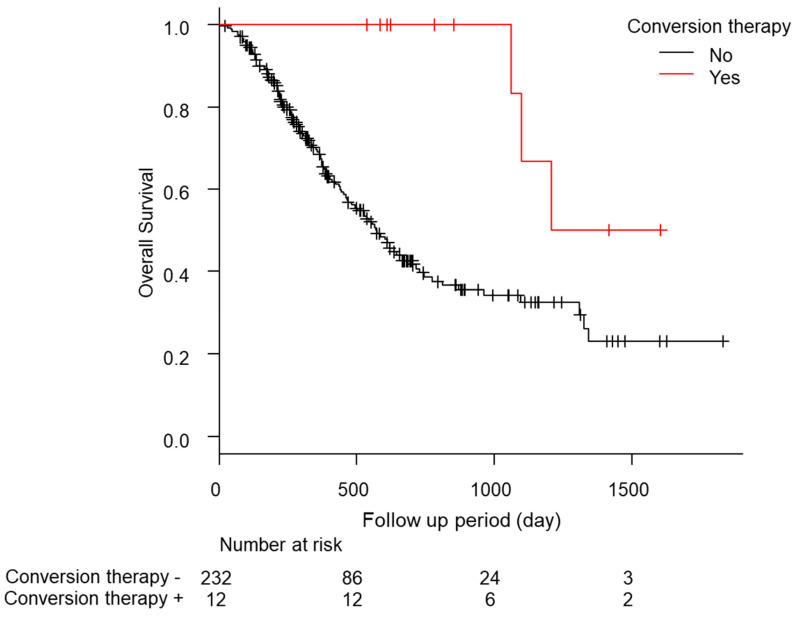
Kaplan-Meier analysis of the overall survival among the patients with conversion cases.

**Figure 3 cancers-15-05221-f003:**
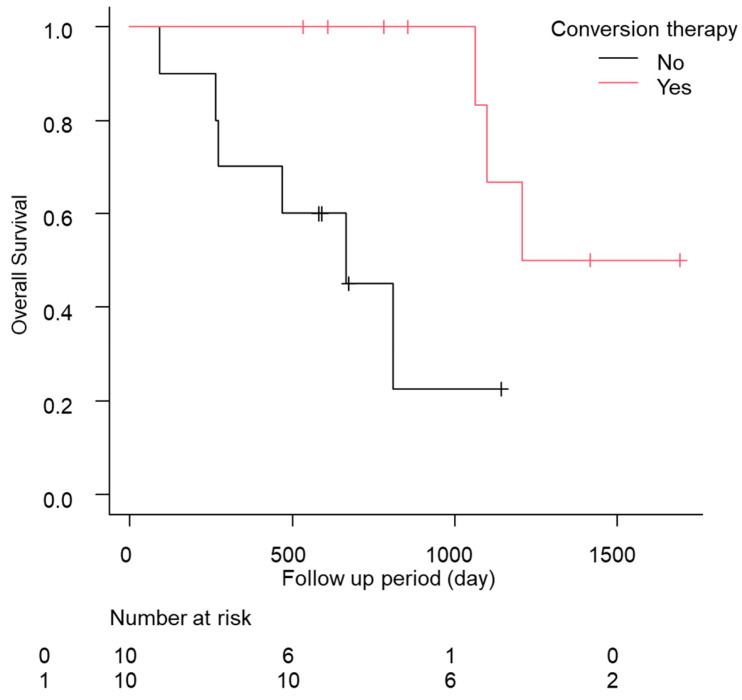
Kaplan–Meier analysis of the overall survival among the patients with conversion cases in propensity score matching.

**Table 1 cancers-15-05221-t001:** Characteristics of patients with unresectable advanced hepatocellular carcinoma treated with lenvatinib and atezolizumab + bevacizumab therapy.

Characteristics	All (n = 244)	Atezo + Bev (n = 113)	LEN (n = 131)	*p*-Value
Age, median [quartiles], (years)	73 [67–79]	73 [68–79]	73 [47–93]	0.45
Sex (male/female), n	196/48	96/17	100/31	0.11
ECOG-PS (0/1/2), n	189/41/14	82/18/13	107/23/1	0.52
Etiology (HBV/HCV/NBNC), n	32/94/118	13/45/55	19/49/63	0.21
Platelets, median [quartiles], (10^4^/μL)	15.2 [10.6–18.7]	15.0 [10.9–18.7]	16.4 [13.5–19.4]	0.59
M2BpGi [qQuartiles] (C.O.I)	1.89 [1.06–4.21]	2.04 [1.06–4.66]	1.44 [0.95–2.41]	0.09
Child–Pugh score (5/6/7/8), n	142/80/16/6	72/33/6/2	70/47/10/4	0.27
mALBI Grade (1/2a/2b/3), n	84/66/90/4	38/32/39/4	46/34/51/0	0.51
Portal vein invasion (absent/present), n	192/52	87/26	105/26	0.64
Extrahepatic spread (absent/present), n	171/73	77/36	94/37	0.67
AFP, median [quartiles] (ng/mL)	87 [8–1013]	82 [8–1229]	54.5 [8–745]	0.83
BCLC stage (B/C), n	127/117	54/59	73/58	0.30
Treatment line (first-line/second-line/third-line/fourth-line/ fifth-line), n	156/51/25/8/4	70/18/13/8/4	86/33/12/0/0	0.01

AFP, alpha-fetoprotein; ALBI, albumin–bilirubin; BCLC, Barcelona Clinic Liver Cancer; ECOG-PS, Eastern Cooperative Oncology Group performance status; M2BpGi, mac-2 binding protein glycosylation isomer; HBV, hepatitis B virus; HCV, hepatitis C virus; NBNC, non-B non-C.

**Table 2 cancers-15-05221-t002:** Characteristics of the patients who were eligible for conversion therapy from lenvatinib or atezolizumab + bevacizumab to conversion therapy.

Characteristics	All (n = 12)	Atezo + Bev (n = 6)	LEN (n = 6)	*p*-Value
Age, median [quartiles], (years)	73 [71–78]	77 [66–78]	72 [71–74]	0.31
Sex (male/female), n	8/4	4/2	4/2	1
ECOG-PS (0/1), n	9/3	4/2	5/1	1
Etiology (HBV/HCV/NBNC), n	0/5/7	0/4/2	0/1/5	0.55
Platelets, median [quartiles], (10^4^/μL)	18.5 [12.1–19.5]	15.2 [9.8–18.7]	18.5 [14.2–21.0]	0.75
M2BpGi [quartiles] (C.O.I)	1.11 [0.97–1.39]	1.73 [0.99–2.45]	1.09 [0.97–1.21]	0.07
Child–Pugh score (5/6), n	9/3	5/1	4/2	1
mALBI Grade (1/2a/2b), n	4/7/1	2/4/0	2/3/1	1
Portal vein invasion (absent/present), n	11/1	5/1	6/0	1
Extrahepatic spread (absent/present), n	12/0	6/0	6/0	1
AFP, median [quartiles] (ng/mL)	13 [5–409]	84 [5–14]	13 [8–1231]	0.93
BCLC stage (B/C), n	11/1	5/1	6/0	1
Treatment line (first-line/second-line/third-line/fourth-line), n	11/0/0/1	5/0/0/1	6/0/0/0	1

HBV, hepatitis B virus; HCV, hepatitis C virus; NBNC, non-B non-C; ECOG-PS, Eastern Cooperative Oncology Group performance status; M2BpGi, mac-2 binding protein glycosylation isomer; ALBI, albumin–bilirubin; AFP, alpha-fetoprotein; BCLC, Barcelona Clinic Liver Cancer; LEN, Lenvatinib; Atezo + Bev, atezolizumab + bevacizumab.

**Table 3 cancers-15-05221-t003:** Univariate analyses of the factors that influenced conversion therapy.

Variables	Category	No. of Patients (%)		Univariate
		Conversion +	Conversion −	*p*-Value
Age, (years)	≥75	5 (42)	97 (42)	1
	<75	7 (58)	135 (58)	
Sex	Male	7 (58)	189 (82)	0.07
	Female	5 (42)	43 (18)	
Etiology	Viral	5 (42)	121 (52)	0.56
	Non-viral	7 (68)	111 (48)	
ECOG PS	1	2 (17)	47 (20)	1
	0	10 (83)	185 (80)	
mALBI Grade 1+2a	Yes	11 (92)	139 (60)	0.03
	No	1 (8)	93 (40)	
Child–Pugh score 5	Yes	9 (75)	134 (58)	0.37
	No	3 (25)	98 (42)	
BCLC stage	B	11 (92)	116 (50)	0.01
	C	1 (8)	116 (50)	
AFP level (ng/mL)	≥400	3 (25)	80 (34)	0.76
	<400	9 (75)	152 (66)	
Treatment line	First-line	11 (92)	145 (62.5)	0.06
	Later-line	1 (8)	87 (27.5)	
Drug therapy	Lenvatinib	6 (50)	125 (54)	0.86
	Atezo + Bev	6 (50)	107 (46)	

AFP, alpha-fetoprotein; Atezo + Bev, atezolizumab + bevacizumab; ECOG PS, Eastern Cooperative Oncology Group performance status.

**Table 4 cancers-15-05221-t004:** Characteristics of tumor factor changes in the conversion cases.

Case	AFP (ng/mL)	Tumor Number	Maximum Tumor Size (mm)	BCLC Stage	RECIST Version 1.1	Treatment Duration (Days)	Treatment Interruption (Days)	Conversion Therapy
1	13→13	5→1	23→15	B→0	PR	379	7	MWA
2	6→6	4→1	51→18	B→0	PR	213	10	MWA
3	1637→5	3→1	73→35	B→A	PR	232	7	Hepatectomy
4	4→4	10→1	61→41	B→A	PR	287	64	Hepatectomy
5	10,950→3660	2→2	42→25	B→A	PR	145	9	MWA
6	11→11	1→1	64→38	A→A	PR	397	209	MWA
7	162→3	1→1	81→41	C→C	PR	168	28	Hepatectomy
8	832→8	5→3	33→13	B→A	PR	210	28	RFA
9	3024→11	3→1	32→11	B→0	PR	126	28	RFA
10	5→5	6→2	28→16	B→A	PR	105	35	MWA
11	268→76	4→1	21→13	B→0	PR	210	28	MWA
12	3→3	5→3	68→23	B→A	PR	441	35	MWA
Median	88→7	4→1	47→25			211	28	

AFP, alpha-fetoprotein; BCLC, Barcelona Clinic Liver Cancer; MWA, microwave ablation; PR, partial response; RECIST, Response Evaluation Criteria in Solid Tumors; RFA, radiofrequency ablation.

**Table 5 cancers-15-05221-t005:** Characteristics of the progress after conversion treatment and post-treatment.

Case	Conversion Therapy	Continuation of Post-Conversion Drug Therapy	Recurrence	Recurrence Pattern	RFS (Days)	After Treatment
1	MWA	+	+	IM	239	MTAs
2	MWA	+	+	IM	308	MWA
3	Hepatectomy	-	+	MC	1357	MWA
4	Hepatectomy	-	+	DM (Bone)	447	MTAs
5	MWA	+	+	DM (Lymph node)	657	MTAs
6	MWA	-	+	Local Recurrence	183	MWA
7	Hepatectomy	-	-	-	748	-
8	RFA	+	-	-	575	-
9	RFA	-	+	IM	167	RFA
10	MWA	-	+	IM	157	TACE
11	MWA	-	+	IM	476	MTAs
12	MWA	+	-	-	68	-
Median					378	

DM, distant metastasis; IM, intrahepatic metastasis; MC, multicentric; MTAs, multi-targeted agents; MWA, microwave ablation; RFA, radiofrequency ablation; RFS, recurrence-free survival; TACE, transcatheter arterial chemoembolization.

## Data Availability

The dataset is available from the corresponding author on reasonable request.

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
