# Peer review of "Clinical Features and Outcomes of Conversion Therapy in Patients with Unresectable Hepatocellular Carcinoma"

_cancers, 2023, doi:10.3390/cancers15215221_

Round 1

Reviewer 1 Report

Comments and Suggestions for Authors

The authors showed that 12 (4.9%) patients who underwent conversion therapy had a favorable prognosis among patients with uHCC treated with LEN and Atezo+Bev, and these patients had good hepatic reserve and function and BCLC-B status.

Overall, the present manuscript is well-written and quite informative but it still has some concerns as indicated below.

Major comments:

1. Patients treated with LEN and Atezo+Bev had the different treatment protocols and schedules, and they had the different antitumor effects after the treatments. The factors associated with the possibility of conversion therapy may not coincide between the two treatments. The authors should compare the baseline characteristics between patients with and without conversion therapy in each treatment group although a few numbers of patients had conversion therapy in each treatment group.

2. The authors defined the indication of conversion therapy in patients who could achieve downstaging of HCC from BCLC stage B or C to A or 0 in the Methods section. However, the authors described that Case 6 achieved downstaging from BCLC stage A to A in Table 4. The authors should modify the definition of conversion therapy.

3. In Figure 1, Case 6 had long interruption period of LEN just before conversion therapy. In Case 6, it seems that LEN was switched to other treatment because LEN was discontinued due to AEs, rather than achieving conversion therapy. It is better that the authors would present the detailed information of the clinical course in Case 6.

4. In Figure 2, the authors showed the OS was significantly longer in patients with conversion therapy than those without conversion therapy. In this comparison, the survival time of patients with conversion therapy included immortal time and the immortal time bias could affect the results of the prolonged survival time of patients with conversion therapy. The authors had better mention the bias in the analysis.

5. In Table 5, eight of the 12 cases relapsed after conversion therapy but Case 3 without recurrence received MWA after conversion therapy. It is better that the authors would present the detailed information of the clinical course after conversion therapy in Case 3.

Minor comments:

1. I would like to receive the detailed information about lesions that recurred after conversion therapy, such as local recurrence of lesions targeted for conversion therapy, local recurrence of lesions targeted for LEN and Atezo+Bev that were not treated with conversion, and new lesions after conversion therapy.

Author Response

Major comments:

  1. Patients treated with LEN and Atezo+Bev had the different treatment protocols and schedules, and they had the different antitumor effects after the treatments. The factors associated with the possibility of conversion therapy may not coincide between the two treatments. The authors should compare the baseline characteristics between patients with and without conversion therapy in each treatment group although a few numbers of patients had conversion therapy in each treatment group.

Thank you for your important remarks. We have conducted a comparative analysis of the Atezo + Bev and LEN groups, considering patients both with and without Conversion treatment. To provide a comprehensive overview of these comparisons, we have incorporated the following text into our manuscript on Page 6, Lines 201-204, along with the inclusion of Figures S3 and S4.

Furthermore, it is noteworthy that within the LEN group, a higher rate of conversions to Conversion therapy was observed in BCLC-B cases (Table S3). Conversely, in the Atezo + Bev group, a significantly greater proportion of conversions to Conversion therapy was observed among patients with mALBI-1+2a (Table S4). 

  1. The authors defined the indication of conversion therapy in patients who could achieve downstaging of HCC from BCLC stage B or C to A or 0 in the Methods section. However, the authors described that Case 6 achieved downstaging from BCLC stage A to A in Table 4. The authors should modify the definition of conversion therapy.

Thank you for your valuable input. As you correctly pointed out, Case 6 is indeed exceptional, and I have made the following additions (Page 3, Lines 116-129)

In most cases, we established specific criteria for initiating conversion therapy in patients who had the potential to achieve downstaging of HCC from BCLC stage B or C to A or 0. The choice of treatment for conversion therapy hinged on several factors, including the patient's general condition, hepatic reserve function, tumor size, tumor number, and the precise localization of HCC. These decisions were made collaboratively, following consultation with our liver surgeon. Ablation procedures were predominantly conducted using microwave ablation or radiofrequency ablation.

Case 6, however, deviated from this typical approach. Despite being classified as a BCLC-A case, we initiated treatment with LEN as an exception. This decision was rooted in the patient's inoperable status, primarily due to an elevated ICG value of 47%. Subsequently, we discontinued LEN therapy due to adverse events and opted for hepatic arterial embolization after the patient's recovery. Following this, microwave ablation therapy was performed as an additional radical treatment.

  1. In Figure 1, Case 6 had long interruption period of LEN just before conversion therapy. In Case 6, it seems that LEN was switched to other treatment because LEN was discontinued due to AEs, rather than achieving conversion therapy. It is better that the authors would present the detailed information of the clinical course in Case 6.

Thank you for pointing this out. We have incorporated the clinical course of Case 6 into the manuscript on Page 3, Lines 123-129, as you suggested.

Case 6, however, deviated from this typical approach. Despite being classified as a BCLC-A case, we initiated treatment with LEN as an exception. This decision was rooted in the patient's inoperable status, primarily due to an elevated ICG value of 47%. Subsequently, we discontinued LEN therapy due to adverse events and opted for hepatic arterial embolization after the patient's recovery. Following this, microwave ablation therapy was performed as an additional radical treatment.

  1. In Figure 2, the authors showed the OS was significantly longer in patients with conversion therapy than those without conversion therapy. In this comparison, the survival time of patients with conversion therapy included immortal time and the immortal time bias could affect the results of the prolonged survival time of patients with conversion therapy. The authors had better mention the bias in the analysis.

Thank you for bringing this to our attention. We have taken your suggestion into account and included the following text in the Discussion section under the limitations. (Page 5, Lines 380-383):

In addition, although the conversion group demonstrated a significant extension in overall survival, it is essential to acknowledge the presence of immortal time within this group. It is plausible that immortal time bias may have influenced the observed survival outcomes in this study.

  1. In Table 5, eight of the 12 cases relapsed after conversion therapy but Case 3 without recurrence received MWA after conversion therapy. It is better that the authors would present the detailed information of the clinical course after conversion therapy in Case 3.

 I am grateful for your remark. In the case of Case 3, it was a recurrence after conversion treatment, which means that the information presented in Table 3 was inaccurate. We have made the necessary revisions to Table 3 to reflect the correct information.

Minor comments:

  1. I would like to receive the detailed information about lesions that recurred after conversion therapy, such as local recurrence of lesions targeted for conversion therapy, local recurrence of lesions targeted for LEN and Atezo+Bev that were not treated with conversion, and new lesions after conversion therapy.

I extend my thanks for your contribution. We have now included the recurrence pattern of post-conversion cases into Table 5.

Reviewer 2 Report

Comments and Suggestions for Authors

Interesting study. The authors should comment on the concept of post-recurrence/progression survival that influence the need of conversion therapy in HCC patients (cite the series PMID: 25085684)

The sample size is very limited so this paper represents rather a case series

Did the authors apply a log-rank test to compare the two KM curves?

The authors should comment on the potential impact of underlying therapy on patient outcome (cite the series PMID: 25974743)

English grammar should be improved

Comments on the Quality of English Language

I recommend minor language revision

Author Response

Reviewer 2

Interesting study. The authors should comment on the concept of post-recurrence/progression survival that influence the need of conversion therapy in HCC patients (cite the series PMID: 25085684)

Thank you for your valuable comment. We have included the following text in the Discussion section and have also added it to the references (Page 5, Lines 372-376):

On the other hand, in patients with hepatocellular carcinoma after radiofrequency ablation, previous studies have highlighted the significance of clinical and tumor parameters evaluated at the time of recurrence, especially the type of recurrence pattern, in influencing survival after recurrence. Therefore, the evaluation at the time of recurrence may hold equal importance in conversion cases.39

  1. Antonio F.; Valentina D.; Matteo A.; Nicola C.; Viviana N.; Alfredo L;. Brian C;. Michele B. Post-recurrence survival in hepatocellular carcinoma after percutaneous radiofrequency ablation. Dig Liver Dis 2014, 11, 1014-9.

The sample size is very limited so this paper represents rather a case series

I am grateful for your noteworthy comment. As you rightly pointed out, the number of conversion cases in our study is indeed limited. However, we considered the 244 cases we reviewed to be a significant sample size given the novelty of the concept of conversion therapy for HCC. The small number of cases achieving conversion is a reflection of the emerging nature of this therapeutic approach. We firmly believe that this study provides essential foundational data that will be instrumental in guiding future prospective studies. Hence, we hope you recognize the importance of presenting this study not just as a case series but as a retrospective clinical study.

Did the authors apply a log-rank test to compare the two KM curves?

Thank you for pointing this out. We have made the following correction in the document:

PFS and OS were analyzed using the Kaplan-Meier curve with the log-rank test. (Page 4, Line 141)

The authors should comment on the potential impact of underlying therapy on patient outcome (cite the series PMID: 25974743)

Thank you for your valuable comment. We have included the following text in the Discussion section and added it to the references. (Page 4, Lines 362-364)

Furthermore, the possibility of the influence of underlying therapy on the post-treatment course has also been reported, and further accumulation of cases is needed in the future.38

  1. Antonio F.; Valentina D.; Nicola C.; Brian C;. Alfredo L;. Michele B. Angiotension receptor blockers improve survival outcomes after radiofrequency ablation in hepatocarcinoma patients. J Gastroenterol Hepatol 2015, 11, 1643-50.

English grammar should be improved

Thank you for pointing this out. We have reviewed and corrected the grammar as a whole.

Round 2

Reviewer 2 Report

Comments and Suggestions for Authors

The authors significantly improved their paper. Thank you!

Just the authors should check carefully the bibliography. The last two references were inappropriately reported (names before surnames). Please amend it.

Author Response

Just the authors should check carefully the bibliography. The last two references were inappropriately reported (names before surnames). Please amend it.

Thank you for pointing this out. We have corrected the references according to your instructions.